# Study on Crystal Growth of Tobermorite Synthesized by Calcium Silicate Slag and Silica Fume

**DOI:** 10.3390/ma16031288

**Published:** 2023-02-02

**Authors:** Zhijie Yang, Chengyang Fang, Yang Jiao, De Zhang, Dong Kang, Kaiyue Wang

**Affiliations:** 1School of Mining and Technology, Inner Mongolia University of Technology, Hohhot 010051, China; 2Key Laboratory of Geological Hazards and Geotechnical Engineering Defense in Sandy and Drought Regions at Universities of Inner Mongolia Autonomous Region, Hohhot 010051, China; 3Inner Mongolia Engineering Research Center of Geological Technology and Geotechnical Engineering, Hohhot 010051, China

**Keywords:** CSS, C-S-H, synthesis time, tobermorite, crystal growth

## Abstract

In order to high-value utilize the secondary solid waste calcium silicate slag (CSS) generated in the process of the extraction of alumina from fly ash, in this paper, tobermorite was synthesized using CSS and silica fume (SF) at different hydrothermal synthesis times. The hydrothermal synthesis was evaluated by means of XRD, SEM, EDS, and micropore analysis, and the results discussed. The results indicate that β-dicalcium silicate, the primary phase in the CSS, partially hydrates at the beginning of hydrothermal synthesis conditions to form mesh-like crystal C-S-H (calcium-rich) and calcium hydroxide. It then reacts with SF to form yarn-like crystal C-S-H (silicon-rich) and then furtherly grows into large flake-like crystal C-S-H (silicon-rich) at 3 h. When the synthesis time is 4 h, β-dicalcium silicate completely hydrates, and crystal C-S-H (calcium-rich) and calcium hydroxide further reacts with large flake-like crystal C-S-H (silicon-rich) to generate medium flake-like tobermorite. With the increase in time, the crystal of hydrothermal synthesis grows in the order of medium flake-like tobermorite, small flake-like tobermorite, strip flake-like tobermorite, fibrous-like tobermorite, and spindle-like tobermorite, and the APV, APD, and SSA show a trend of decreasing first, then increasing, and then decreasing. Meanwhile, strip flake-like tobermorite with a higher average pore volume (APV), average pore diameter (APD), and specific surface area (SSA) can be synthesized at 6 h.

## 1. Introduction

There are many types of calcium silicate hydrate minerals in the CaO-SiO_2_-H_2_O system. There are more than 30 kinds of calcium silicate hydrate minerals that can stably exist, including 5CaO·6SiO_2_·3H_2_O, CaO·SiO_2_·H_2_O, 2CaO·SiO_2_·1.17H_2_O, 2CaO·3SiO_2_·2.5H_2_O, CaO·2SiO_2_·2H_2_O, 3CaO·2SiO_2_·3H_2_O, tobermorite, and xonotlite [1]. Different calcium silicates, as non-toxic and non-hazardous non-metallic inorganic functional materials, are attracting more and more attention due to their broad application prospects [2,3,4,5]. In particular, tobermorite has been widely used in building materials, and in the chemical and metallurgical industries due to its good heat resistance and thermal insulation properties [6]. As a main mineral of cement hydration, there are three different kinds of crystal tobermorite according to the interlayer water content, including 9 Å tobermorite, 11 Å tobermorite, and 14 Å tobermorite. The 14 Å tobermorite is the most hydrous one. The tobermorite under ambient conditions is 11 Å tobermorite [7,8].

In recent years, calcium silicate hydrate materials were prepared by different methods, such as hydrothermal synthesis [9], aqueous solution reaction [10], and Ca(NO_3_)_2_·4H_2_O and Na_2_SiO_3_·5H_2_O complex decomposition. However, the CaO and SiO_2_ mechanochemical method [11] has become one of the research hotspots for non-metallic synthesis materials. CSS is a new solid waste generated during the extraction of alumina from fly ash, and 2 tons of CSS is discharged when 1 ton of alumina is produced from 2.5 tons of fly ash [12]. Therefore, how to treat and utilize CSS has become an urgent problem in the recycling industry chain during the extraction of alumina from fly ash. Because CSS has good hydration properties, and its main chemical composition is CaO and SiO_2_, it is a suitable raw material for synthesizing calcium silicate hydrate materials. This research was carried out on the hydrothermal synthesis of tobermorite using CSS and SF as the primary calcium silicon source [13,14]. Lucie Galvánková, from the Brno University of Technology, Czech Republic, synthesized needle-like tobermorite using SF and lime under a C/S moral ratio of 0.83 [15]. Furthermore, Cristian Biagioni, from Italy, provided a naming method for different tobermorite crystals [16]. Yinusa Daniel Lamidi, of the Federal Polytechnic, Nigeria, used shell, waste glass, and other solid waste to synthesize tobermorite [17]. As a product of cement hydration, tobermorite has great influence on the strength and durability of cement [18]. 

In the past, the research on tobermorite mainly focused on the influence on cement performance as a cement hydration product, while the high-purity tobermorite synthesized in recent years has been widely used as a functional material. The author found that these different crystal forms of tobermorite have a wide range of utilizations and high utilization value. Porous tobermorite, with a specific surface area of 150–200 m^2^/g, can be used as paper filler and cement reinforcement agent. Tobermorite, with a specific surface area of less than 150 m^2^/g, can be used as sewage treatment and an air purifying agent, and as rubber and plastic filler. Fibrous tobermorite can be used to produce calcium silicate insulation materials, and the author has confirmed these through an industrial experiment and found that different crystal forms of tobermorite can not only significantly improve the material’s performance, but also have significant economic benefits. However, in the process of industrial production, it was found that the structure, particle size, porosity, fiber length, and diameter ratio of tobermorite are often different, which makes it difficult to continuously and stably produce tobermorite products that meet the performance requirements of different industries. The reason is that the crystal growth process of tobermorite synthesized by CSS and SF has not been clearly understood, and is not conducive for crystal control or for the determination of the synthesis process at present.

Some scholars have also carried out relevant research on the crystal growth of C-S-H synthesized by hydrothermal method. Fang, Q. has studied the crystallization behavior of C-S-H synthesized by fly ash desilication solution in strong alkaline environment through crystal structure and kinetics. The results showed that the synthesis temperature is the main factor affecting the crystallization of C-S-H, and the strong alkaline environment will promote to form fibrous C-S-H [19]. Pengxu, C. synthesized fibrous tobermorite with fly ash and lime through alkali strengthening at 200 °C. Only when the synthesis time is extended to 3–7 h, fibrous tobermorite with a length of 2–3 μm and a diameter of 0.01–0.5 μm can be produced. However, when the synthesis temperature is increased to 220 °C, the tobermorite fiber will be transformed into spherical constructions [20]. Nguyen, D.V. studied the effect of hydrothermal synthesis conditions on the crystal growth of xonotlite. The results showed that the order of formation of hydrated calcium silicate was low crystallization degree C-S-H, crystalline tobermorite, xonotlite [21]. However, as a new type of solid waste, there is little research on the synthesis of C-S-H from CSS, especially in the hydrothermal synthesis of tobermorite.

Therefore, to study the synthesis reaction mechanism and crystal growth process of tobermorite, this study carried out dynamic hydrothermal synthesis experiments of tobermorite using CSS and SF under different hydrothermal synthesis times. The purpose was to provide a theoretical basis for the crystal control of tobermorite using CSS and SF. The tobermorite with different crystal forms can not only be widely used in paper, plastics, rubber fillers, and thermal insulation materials, but also solve the utilization problem of CSS and promote the development of a circular economy industry chain of the extraction of alumina from fly ash.

## 2. Materials and Methods

### 2.1. Materials

The primary raw materials used in this experiment included CSS, SF, and distilled water. The CSS was sourced from the Datang International Renewable Resources Co., Ltd. (Hohhot, China), the world’s first commercial operation production line for extraction aluminum from fly ash. The SF came from Inner Mongolia Erdos Electric Power Metallurgy Co., Ltd. (Erdos, China).

The chemical compositions and phase compositions of raw materials were analyzed using X-ray fluorescence (Shimadzu, Kyoto, Japan, XRF-1800) and X-ray diffractometer, and the results are shown in Table 1. The primary phases of CSS are β-dicalcium silicate (β-2CaO·SiO_2_) and calcite (CaCO_3_). The main phases of SF are glass phase.

### 2.2. Experimental Method

The CSS was dried in the oven at 110 °C until constant weight and then milled in a ball mill until particle size less than 300 mesh. According to the C/S molar ratio of 0.9, the liquid–solid ratio of 15:1, 100 g CSS and 48.3 g SF were weighed for each tobermorite synthesis experiment, and 2224.5 g distilled water was added. After mixing, the mixture was placed in a 5 L high-pressure reactor (GHS-5 L). The tobermorite hydrothermal synthesis experiments were executed according to a stirring rate of 300 r/min and a heating rate of 20 °C/min until 240 °C and kept at a constant temperature for the corresponding hydrothermal synthesis time. After the end of the constant temperature, the high-pressure reactor was naturally cooled to room temperature. Then the hydrothermal synthesis was removed from the high-pressure reactor, dehydrated through a Circulating water vacuum filter (SHZ-D(Ⅲ), Kerui Instrument Co., LTD, Gonyi, China), and dried in the oven at 110 °C until constant weight. 

Finally, the phase composition of hydrothermal synthesis was tested by X-ray diffraction (PANalytical, Almelo, Netherlands, X’Pert Powder 3) under condition of Cu target, 40 kV, scanning range 10°~100°, step 0.02°, and the XRD test result was analyzed through X’Pert Highscore software (2.0 version) [22]. The micromorphology of hydrothermal synthesis was tested by scanning electron microscopy (Hitachi S-4800, Hitachi, Tokyo, Japan), and SSA and pore size distribution analyzer (V-Sorrb 2800TP, Gold App Instruments Corporation, Beiijing, China) were used to test and analyze the APV, APD, and SSA of the hydrothermal synthesis. The synthesis time gradually increased from 1 h until 8 h, and one hydrothermal synthesis experiment was executed when synthesis time increased every hour, a total of 8 experiments.

## 3. Discussion

### 3.1. Phase Evolution Analysis of Hydrothermal Synthesis

XRD analysis of hydrothermal synthesis synthesized using CSS and SF at different hydrothermal synthesis times is shown in Figure 1. When the hydrothermal synthesis time was at 1 h, the primary phases of hydrothermal synthesis were crystal C-S-H (cal-cium-rich), unreacted β-dicalcium silicate, and calcite. However, when the hydro-thermal synthesis time reached to 2 and 3 h, the primary phase of the hydrothermal synthesis was crystal C-S-H (silica-rich). Finally, when the hydrothermal synthesis time continued to increase to more than 4 h, the primary phase of the hydrothermal synthesis was 11 Å tobermorite (5CaO·6SiO_2_·5H_2_O).

The XRD analysis of hydrothermal synthesis synthesized using CSS and SF at different hydrothermal synthesis times is shown in Figure 1. When the hydrothermal synthesis time was at 1 h, the primary phases of hydrothermal synthesis were crystal C-S-H (calcium-rich), unreacted β-dicalcium silicate, and calcite. However, when the hydrothermal synthesis time reached 2 and 3 h, the primary phase of the hydrothermal synthesis was crystal C-S-H (silica-rich). Finally, when the hydrothermal synthesis time continued to increase to more than 4 h, the primary phase of the hydrothermal synthesis was 11 Å tobermorite (5CaO·6SiO_2_·5H_2_O).

The hydrothermal synthesis reaction mechanism of tobermorite using CSS and SF can be inferred from the evolution process of the main phase of the hydrothermal synthesis and EDS analysis results of different micromorphology phases in Table 2. Because β-dicalcium silicate is a kind of slow hydration mineral, Mohammadreza researched the dissolution of β-C2S cement clinker, and the results showed that the total time taken to dissolve the whole β-C2S crystal was between 1400 and 2400 s with a mesoscopic forward dissolution rate of 4.15 × 10^−12^ mol/(s.cm^2^) [20]. The hydration degree of β-dicalcium silicate is only 10.3% under the condition of room temperature after hydrating for 28 days [21]. Thus, β-dicalcium silicate, as a primary phase in the CSS, first is partially hydrated under hydrothermal synthesis conditions to generate crystal C-S-H (calcium-rich) and calcium hydroxide, as shown in Equation (1), and crystal C-S-H (calcium-rich), as shown in region 1 of Table 2. Furthermore, calcium hydroxide dissociates in water to form Ca(H_2_O_5_)(OH)^+^, which makes the whole system become strongly alkaline. Therefore, under the polarization of OH^−^, the high degree of polymerization of Si-O in SF is depolymerized to form H_2_SiO_4_^2−^ dominated by Q^0^ structural units. Finally, H_2_SiO_4_^2−^ reacts with crystal C-S-H (calcium-rich) and Ca(H_2_O_5_)(OH)^+^ to form crystal C-S-H (silicon-rich), as shown in Equation (2). When the hydrothermal synthesis time increases to more than 4 h, all the β-calcium silicate hydrates and generates more crystal C-S-H (calcium-rich) and calcium hydroxide. They then react with the crystal C-S-H (silicon-rich) generated previously to generate 11 Å tobermorite.
(1)$$2\mathrm{CaO}\cdot {\mathrm{SiO}}_{2}+H_{2}O\to C-S-H\left(\mathrm{calcium}-\mathrm{rich}\right)+\mathrm{Ca}{\left(\mathrm{OH}\right)}_{2}$$
(2)$$C-S-H\left(\mathrm{calcium}-\mathrm{rich}\right)+\mathrm{Ca}\left(H_{2}O_{5}\right){\left(\mathrm{OH}\right)}^{+}+H_{2}{\mathrm{SiO}}_{4}^{2-}\to C-S-H\left(\mathrm{silicon}-\mathrm{rich}\right)$$

According to the above analysis results, at a C/S molar ratio of 0.9, the synthesis temperature is 240 °C, and the main phases in the growth process are crystal C-S-H (calcium-rich), crystal C-S-H (silicon-rich), and 11 Å tobermorite. At the same time, through the XRD analysis of hydrothermal synthesis, it can be proved that the reaction activity of CSS and SF is high under high temperature and high alkaline environment, and sufficient Ca (H_2_O_5_) (OH)^+^ and H_2_SiO_4_^2−^ will be formed in the solution, which can ensure the formation and transformation of corresponding crystalline calcium silicate minerals. This shows that CSS is a good raw material for synthesizing calcium silicate minerals from the perspective of chemical reaction.

### 3.2. Micromorphology Evolution Analysis of Hydrothermal Synthesis

Hitachi S-4800 was used to analyze the micromorphology growth process of the hydrothermal synthesis of CSS and SF, and EDS analysis was also carried out in some special regions. The results are combined with the above XRD analysis results and shown in Figure 2. In addition, the phase with different micromorphology was deduced, with the results shown in Table 2.

It can be seen from Figure 2 that the micromorphology of the hydrothermal synthesis of CSS and SF changed significantly with an increase in the hydrothermal synthesis time from mesh-like to yarn-like, flake-like, fibrous-like, and spindle-like. Combined with the analysis in Table 2, the mesh-like crystal C-S-H (calcium-rich) is mainly generated at 1 h, as shown in Figure 2a. With an increase in synthesis time, the mesh-like crystal C-S-H (calcium-rich) continues to grow more dendrites, making the mesh smaller and smaller, and then the yarn-like crystal C-S-H (silicon-rich) is generated, as shown in Figure 2b. With the continuing increase in the hydrothermal synthesis time, the crystal C-S-H (calcium-rich) reacts with H_2_SiO_4_^2−^ anions to further generate large flake-like crystal C-S-H (silicon-rich), as shown in Figure 2c. When the synthesis time increases to more than 4 h, the large flake-like crystal C-S-H (silicon-rich) undergoes evident crystal evolution and gradually forms medium flake-like and small flake-like 11 Å tobermorite with smaller grain sizes, as shown in Figure 2d,e. Up to the synthesis time of 6 h, the strip flake-like 11 Å tobermorite is formed, as shown in Figure 2f. The strip flake-like 11 Å tobermorite continues to evolve with the extension of synthesis time, forming fibrous-like 11 Å at 7 h, as shown in Figure 2g, and fibrous-like 11 Å tobermorite continues to evolve and form spindle-like 11 Å tobermorite at 8 h, as shown in Figure 2h.

In view of the above analysis, with an increase in synthesis time, the micromorphology evolution process of the hydrothermal synthesis synthesized by CSS and SF is mesh-like, yarn-like, large flake-like, medium flake-like, small flake-like, strip flake-like, fibrous-like, and spindle-like. And through the micromorphology analysis of hydrothermal synthesis, it is found that the calcium silicate minerals with the same phase have different micromorphology, especially the micromorphology of 11 Å tobermorite is more diverse. On the one hand it confirms that 11 Å tobermorite has a wide range of applications in different fields, but on the other hand, it also confirms that the microstructure of 11 Å tobermorite is not easy to accurately control, so it is necessary to further study the synthesis mechanism.

### 3.3. Micropore Parameters Evolution Analysis of Hydrothermal Synthesis

The micropores was analyzed for the hydrothermal synthesis synthesized at different synthesis times, shown as in Figure 3. The APV, APD, and SSA of the hydrothermal synthesis decrease first, then increase, and then decrease again with the increase of the synthesis time. In view of analysis results of XRD and SEM, it can be seen that the hydrothermal synthesis is the mesh-like crystal C-S-H (calcium-rich) at the synthesis time of 1 h. Since this crystal structure takes on a porous, it has a high APV, APD, and SSA. With the increase of synthesis time, the crystal of hydrothermal synthesis gradually evolved into yarn-like and flake-like, and the grain size became smaller and smaller, resulting in the gradual increase of the APV, APD, and SSA of the hydrothermal synthesis. At the synthesis time of 6 h, a strip-like 11 Å tobermorite with high APV, APD, and SSA is generated. When the synthesis time is extended to more than 7 h, the hydrothermal synthesis becomes fibrous-like tobermorite-11 Å and spindle-like 11 Å tobermorite. In contrast, the fibrous-like 11 Å tobermorite and spindle-like 11 Å tobermorite have a higher length-diameter ratio and smaller pores between grains, so their APV, APD, and SSA are significantly decreased.

The micropores were analyzed for the hydrothermal synthesis synthesized at different synthesis times, as shown in Figure 3. The APV, APD, and SSA of the hydrothermal synthesis decrease first, then increase, and then decrease again with the increase in the synthesis time. In view of the analysis results of XRD and SEM, it can be seen that the hydrothermal synthesis is the mesh-like crystal C-S-H (calcium-rich) at the synthesis time of 1 h. Since this crystal structure takes on a porous quality, it has a high APV, APD, and SSA. With the increase in synthesis time, the crystal of hydrothermal synthesis gradually evolves into being yarn-like and flake-like, and the grain size becomes smaller and smaller, resulting in the gradual increase in the APV, APD, and SSA of the hydrothermal synthesis. At the synthesis time of 6 h, a strip-like 11 Å tobermorite with a high APV, APD, and SSA is generated. When the synthesis time is extended to more than 7 h, the hydrothermal synthesis becomes fibrous-like tobermorite-11 Å and spindle-like 11 Å tobermorite. In contrast, the fibrous-like 11 Å tobermorite and spindle-like 11 Å tobermorite have a higher length–diameter ratio and smaller pores between grains, so their APV, APD, and SSA are significantly decreased.

In view of above analysis, the APV, APD, and SSA of hydrothermal synthesis initially decrease with the increase in synthesis time, then increase and decrease again. The APV, APD, and SSA are closely related to the micromorphology of hydrothermal synthesis. To synthesize 11 Å tobermorite with a high APV, APD, and SSA, the hydrothermal synthesis time of CSS and SF should be controlled at 6 h. On the other hand, to prepare fibrous-like 11 Å tobermorite, the hydrothermal synthesis time of CSS and SF should be controlled at 7 h.

## 4. Conclusions

In this paper, the synthesis experiments of tobermorite were executed using CSS and SF under different synthesis times. The reaction mechanism and crystal growth of tobermorite were determined through XRD, SEM, EDS, and micropore analysis. The conclusions are as follows:(1)The reaction mechanism for the hydrothermal synthesis of tobermorite using CSS and SF is as follows. The β-dicalcium silicate, the primary phase in the CSS, first partially hydrates under hydrothermal synthesis conditions to form crystal C-S-H (calcium-rich) and calcium hydroxide. The calcium hydroxide then dissociates in water to form Ca(H_2_O_5_)(OH)^+^, which makes the whole system become strongly alkaline. Therefore, under the polarization of OH^−^, the high degree of polymerization of Si-O in SF is depolymerized to form an H_2_SiO_4_^2−^ anion dominated by Q^0^ structural units. Then H_2_SiO_4_^2−^ reacts with crystal C-S-H (calcium-rich) and Ca(H_2_O_5_)(OH)^+^ to form crystal C-S-H (silicon-rich). When the synthesis time increases to more than 4 h, all the β-calcium silicate hydrates and generates more crystal C-S-H (calcium-rich) and calcium hydroxide. This is followed by their reaction with crystal C-S-H (silicon-rich) generated previously, resulting in the 11 Å tobermorite phase.(2)When the C/S molar ratio is 0.9, the synthesis temperature reaches 240 °C. With an increase in the hydrothermal synthesis time, the crystal growth process starts with mesh-like crystal C-S-H (calcium-rich), followed by yarn-like crystal C-S-H (silicon-rich), large flake-like crystal C-S-H (silicon-rich), medium flake-like tobermorite-11 Å, small flake-like 11 Å tobermorite, strip flake-like 11 Å tobermorite, fibrous-like 11 Å tobermorite, and spindle-like 11 Å tobermorite.(3)With an increase in synthesis time, the APV, APD, and SSA of hydrothermal synthesis decrease first, then increase, and then decrease again. The 11 Å tobermorite with a high APV, APD, and SSA can be synthesized by CSS and SF by controlling the synthesis time at 6 h. To prepare fibrous-like 11 Å tobermorite, the hydrothermal synthesis time should be controlled at 7 h.

## Figures and Tables

**Figure 1 materials-16-01288-f001:**
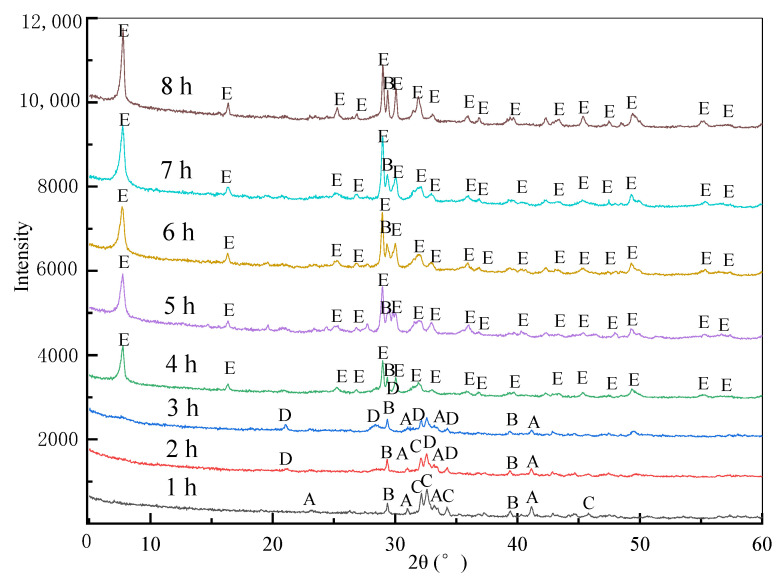
XRD patterns of hydrothermal synthesis prepared by calcium silicate slag and silica fume under different hydrothermal synthesis times. A, β-dicalcium silicate (β-2CaO·SiO_2_); B, calcite (CaCO_3_); C, crystal C-S-H (calcium-rich); D, crystal C-S-H (silica-rich); E, 11 Å tobermorite (5CaO·6SiO_2_·5H_2_O).

**Figure 2 materials-16-01288-f002:**
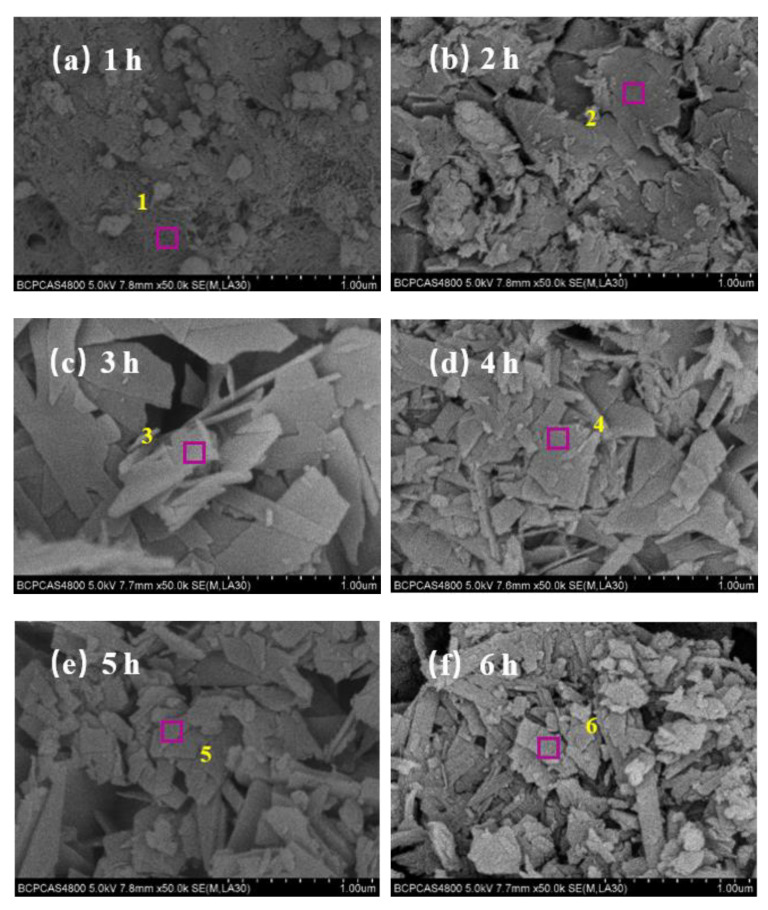
SEM images of hydrothermal synthesis at different hydrothermal synthesis times. (**a**) 1 h, (**b**) 2 h, (**c**) 3 h, (**d**) 4 h, (**e**) 5 h, (**f**) 6 h, (**g**) 7 h, (**h**) 8 h. The pink squares in the figure represent EDS region, while yellow numbers indicate EDS region serial number.

**Figure 3 materials-16-01288-f003:**
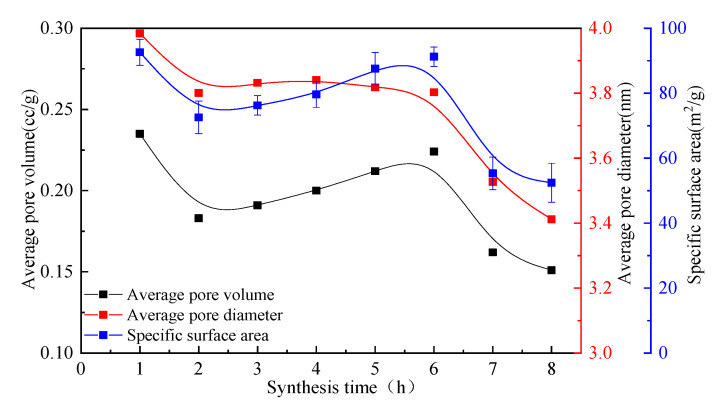
Change in APV, APD, and SSA of the hydrothermal synthesis with an increase in hydrothermal synthesis time.

**Table 1 materials-16-01288-t001:** Content of chemical composition for raw materials (mass fraction %) [13,14].

Chemical Composition	SiO_2_	Fe_2_O_3_	Al_2_O_3_	CaO	MgO	Na_2_O	K_2_O	SO_3_	P_2_O_5_	F	Cl
CSS	29.26	2.53	5.30	55.50	3.61	2.58	0.36	0.73	0.14	-	0.42
SF	72.15	1.16	0.59	7.50	7.43	0.68	4.20	3.17	0.56	1.74	0.82

**Table 2 materials-16-01288-t002:** EDS analysis and corresponding phase of hydrothermal synthesis at different hydrothermal synthesis times.

Region	Micromorphology	Atomic Molar Ratio (%)							Corresponding Phase
		O	Al	Si	Ca	Na	Mg	Fe	
1	Mesh-like	59.37	0.34	14.56	25.32	0.13	0.24	0.04	Crystal C-S-H (calcium-rich)
2	Yarn-like	62.51	0.25	22.37	14.25	0.22	0.32	0.08	Crystal C-S-H (silicon-rich)
3	Large flake-like	60.89	0.37	24.69	13.73	0.16	0.42	0.11	Crystal C-S-H (silicon-rich)
4	Medium flake-like	65.21	0.36	18.25	15.42	0.11	0.52	0.13	11 Å Tobermorite
5	Small flake-like	63.36	0.52	17.22	16.02	0.35	0.81	0.32	11 Å Tobermorite
6	Strip flake-like	64.72	0.33	18.95	15.23	0.12	0.44	0.21	11 Å Tobermorite
7	Fibrous-like	65.17	0.36	18.25	15.42	0.15	0.52	0.13	11 Å Tobermorite
8	Spindle-like	66.03	0.26	18.21	14.83	0.24	0.22	0.21	11 Å Tobermorite

## Data Availability

The data presented in this study are available on request from the corresponding author. When the project was carried out, there was no obligation to make the data publicly available.

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
