# Peer review of "Study on Crystal Growth of Tobermorite Synthesized by Calcium Silicate Slag and Silica Fume"

_materials, 2023, doi:10.3390/ma16031288_

Round 1

Reviewer 1 Report

Following are the suggestions for the authors.

1-      Please use the short keywords for the paper.

2-      Author use different abbreviation at different places, which confused the reader, Please provide the list of the abbreviation, please use in the start.

3-      Abstract could be more informative by providing results. Please include some results at the end of the abstract.

4-      Research gaps should be highlighted more clearly and future applications of this study should be added. Please make the link between the advances images used in this paper with the material properties.

5-      The introduction needs to be more emphasized on the research work with a detailed explanation of the whole process considering past, present and future scope.

Author Response

Author`s reply to the review report (Reviewer 1) Round 1

1.Please use the short keywords for the paper.

Response: Hydrothermal synthesis time in keywords has been changed synthesis time.

2.Author use different abbreviation at different places, which confused the reader, Please provide the list of the abbreviation, please use in the start.

Response:

The calcium silicate slag has been abbreviated to CSS from title start to full text.

The silica fume has been abbreviated to SF from title start to full text.

The average pore volume has been abbreviated to APV in the full text.

The average pore diameter has been abbreviated to APD in the full text.

The specific surface area has been abbreviated to SSA in the full text.

3.Abstract could be more informative by providing results. Please include some results at the end of the abstract.

Response:

The abstract has been revised and some results was supplied in abstract.

4.Research gaps should be highlighted more clearly and future applications of this study should be added. Please make the link between the advances images used in this paper with the material properties.

Response: The future applications of tobermorite has been supplemented in introduction.

5.The introduction needs to be more emphasized on the research work with a detailed explanation of the whole process considering past, present and future scope.

Response: It has been supplemented in introduction.

Reviewer 2 Report

Although the meaning of the manuscript can be gleaned with some effort, there are many parts of it which require attention as to their presentation, syntax and grammar. I mention some indicative ones on page 1, below:

·          There are numerous instances of run-on sentences and random comma placement, often confusing the primary points of the sentence. Interestingly, there are also multiple sentences with an evident lack of punctuation! The first sentence of the Abstract is a prime example of this.

·          Page 1, lines 9-10 “…tobermorite was discussed by means 9 of XRD, SEM, EDS, and micropore analysis.” Should be rephrased, as the result of the analytical methods are discussed, not the methods themselves, i.e. “…tobermorite was evaluated by means 9 of XRD, SEM, EDS, micropore analysis, and the results discussed.

·          Page 1, line 11 “…first partially hydrates at the beginning of hydrothermal synthesis…” please use either “first” or “beginning” using both is redundant.

·          Page 1, lines 29-30 “…have attracted more and more attention…” should read “…are attracting more and more attention…”

·          Page 1, line 31 “…used in building materials and chemical and metallurgical…” should read “…used in building materials, chemical and metallurgical…

·          Page 1, lines 33-34 “…materials by the hydrothermal synthesis method…” should read “…materials by hydrothermal synthesis…” the term “method” re-appears another 3 times in the following 2 rows and is completely redundant. The entire sentence in itself is completely incoherent.

·          Page 1, line 37 “The CSS is as a new solid waste…” should read CSS is as a new solid waste…”

·          Page 1, line 37 “…generated in extraction alumina from…”please rephrase to “…generated during the extraction of alumina from…”

There are many more of these issues throughout the text and I’d ask the authors to carefully proofread their manuscript again and consult a colleague versed in scientific English writing.

Please refrain from abbreviating text in the article title.

Also, keywords are used to assist search engines to find the article, as such, is seems redundant to use keywords apparent in the title.

Concerning the “Materials and Methods” and “Results” section of the manuscript. She entire approach and setup of this study is the same as several previous ones of the author group e.g. Materials 15(4),1620, Journal of Materials Research and Technology 15, pp. 4185-4192. Some parts are actually identical e.g. Table 1 of this study is transferred from a previously published work by the authors (i.e. Journal of Materials Research and Technology 15, pp. 4185-4192). In such cases authors should simply refer to the use of materials presented in the former study. The only difference of this study to the latter ones, is the evaluation of a new parameter (i.e. time) compared to temperature or C/S molar ratio, considered in the previous studies. Since no new methods or concepts approaches are introduced, I would consider this a minor contribution towards the current state of the art and label this a “case study” rather than a research article.

Author Response

Author`s reply to the review report 2 (Reviewer 2) Round 1

Although the meaning of the manuscript can be gleaned with some effort, there are many parts of it which require attention as to their presentation, syntax and grammar. I mention some indicative ones on page 1, below:

There are numerous instances of run-on sentences and random comma placement, often confusing the primary points of the sentence. Interestingly, there are also multiple sentences with an evident lack of punctuation! The first sentence of the Abstract is a prime example of this.

1. Page 1, lines 9-10 “…tobermorite was discussed by means 9 of XRD, SEM, EDS, and micropore analysis.” Should be rephrased, as the result of the analytical methods are discussed, not the methods themselves, i.e. “…tobermorite was evaluated by means 9 of XRD, SEM, EDS, micropore analysis, and the results discussed.”

Response: The hydrothermal synthesis was evaluated by means of XRD, SEM, EDS, micropore analysis, and the results discussed.

2. Page 1, line 11 “…first partially hydrates at the beginning of hydrothermal synthesis…” please use either “first” or “beginning” using both is redundant.

Response: The first is redundant.

3. Page 1, lines 29-30 “…have attracted more and more attention…” should read “…are attracting more and more attention…”

Response: It has been revised.

4. Page 1, line 31 “…used in building materials and chemical and metallurgical…” should read “…used in building materials, chemical and metallurgical…”

 Response: It has been revised.

5.Page 1, lines 33-34 “…materials by the hydrothermal synthesis method…” should read “…materials by hydrothermal synthesis…” the term “method” re-appears another 3 times in the following 2 rows and is completely redundant. The entire sentence in itself is completely incoherent.

Response: It has been revised.

In recent years, the calcium silicate hydrate materials was prepared by different method, such as hydrothermal synthesis [7], aqueous solution reaction [8], Ca(NO3)2·4H2O and Na2SiO3·5H2O complex decomposition. But CaO and SiO2 mechanochemical method [9] has become one of the research hotspots for non-metallic synthesis materials.

Response: It has been revised.

7. Page 1, line 37 “The CSS is as a new solid waste…” should read “CSS is as a new solid waste…”

Response: It has been revised.

8. Page 1, line 37 “…generated in extraction alumina from….”please rephrase to “…generated during the extraction of alumina from…”

Response: It has been revised.

9.There are many more of these issues throughout the text and I’d ask the authors to carefully proofread their manuscript again and consult a colleague versed in scientific English writing.

Response:The paper had been proofread and edited by a native English-speaking reviewer from Cambridge Proofreading Worldwide LLC, which is an institution specialized in proofread English.

10.Please refrain from abbreviating text in the article title.

Response:Abbreviations have been removed from the title.

11.Also, keywords are used to assist search engines to find the article, as such, is seems redundant to use keywords apparent in the title.

Response:The keywords have been replaced.

Reviewer 3 Report

The submitted manuscript is about using calcium silicate slag (CSS) and silica fume (SF) to synthesize tobermorite at different hydrothermal synthesis time. It can be accepted for publication after reasonable explanation for this major revision. I kindly ask authors to prepare a response letter point-by-point rebuttal and must be subjected to the manuscript as well, considering the following comments with sufficient explanations.

1)     The introduction is very short and must be reported comprehensively. What is tobermorite and ad how many types of tobermorite are there? How about the mechanical properties of them form microscale point of view and how about other CSH phases are generated during hydration of cement? The following literatures are suggested to use for reporting this part (https://doi.org/10.1021/acs.jpcc.8b11920https://doi.org/10.1021/acs.jpcc.1c10151).

2)    As reported in the manuscript, the primary phase in CSS is C2S, and according to the following recent publication (https://doi.org/10.3390/ma15196716), the dissolution time of C2S is reported with the contribution of crystal defects on the borders along the Z axis, the total dissolution time of the whole β-C2S crystal was computed between 3 and 4.2 s with mesoscopic forward dissolution rate of 3.30 × 10−9 mol/(s.cm2). However, by considering a perfect (semi-infinitive) crystal using periodic boundary conditions (PBCs) along the Z and Y axes, the total time taken to dissolve the whole β-C2S crystal was between 1400 and 2400 s with mesoscopic forward dissolution rate of 4.15 × 10−12 mol/(s.cm2). According to the reported results how you evaluate rapid dissolution for your samples as long as there is no contribution of C3S in the system and C2S is responsible for late strength of concrete. The results from KMC must be reported in your manuscript and also your reason to make it clear for the readers.

3)    Moreover, as reported in the above literature (https://doi.org/10.3390/ma15196716), the compressive strength of concrete made from pure belite clinker (32.5 MPa) is almost half of pure alite clinker (58.4 MPa). Therefore, how you explain this disadvantage as used in your samples.

4)    Caption from Figure 1 is strange and must be modified (Figure 1. …..!). Moreover, there is not any vertical axes with the unit (it must be intensity). 

5)     It must be 11 A tobermorite not tobermorite 11 A.

6)    How the peaks have been distinguished for Figure 1 (is it coming from the literatures or have been analysed by you(any software) )? In case it is coming from the literatures, they must be cited.

7)    XRD measurement technique and its characterizations must be added to the methodology. 

8)    Please make more comprehensive conclusion as in the revised version the following points must be included; materials and methods, the significant of this study, the scope of the effort, the procedures used to execute the work, and the major findings.

 Best regards,

Author Response

Author`s reply to the review report (Reviewer 3) Round 2 

1.The introduction is very short and must be reported comprehensively. What is tobermorite and ad how many types of tobermorite are there? How about the mechanical properties of them form microscale point of view and how about other CSH phases are generated during hydration of cement? The following literatures are suggested to use for reporting this part (https://doi.org/10.1021/acs.jpcc.8b11920, https://doi.org/10.1021/acs.jpcc.1c10151).

Response:These two articles have been referred and are used as references.

2.As reported in the manuscript, the primary phase in CSS is C2S, and according to the following recent publication (https://doi.org/10.3390/ma15196716), the dissolution time of C2S is reported with the contribution of crystal defects on the borders along the Z axis, the total dissolution time of the whole β-C2S crystal was computed between 3 and 4.2 s with mesoscopic forward dissolution rate of 3.30 × 10−9 mol/(s.cm2). However, by considering a perfect (semi-infinitive) crystal using periodic boundary conditions (PBCs) along the Z and Y axes, the total time taken to dissolve the whole β-C2S crystal was between 1400 and 2400 s with mesoscopic forward dissolution rate of 4.15 × 10−12 mol/(s.cm2). According to the reported results how you evaluate rapid dissolution for your samples as long as there is no contribution of C3S in the system and C2S is responsible for late strength of concrete. The results from KMC must be reported in your manuscript and also your reason to make it clear for the readers.

Response:The article has been referred and are used as references.

3) Moreover, as reported in the above literature (https://doi.org/10.3390/ma15196716), the compressive strength of concrete made from pure belite clinker (32.5 MPa) is almost half of pure alite clinker (58.4 MPa). Therefore, how you explain this disadvantage as used in your samples.

Response:Tobermorite synthesized in this paper is not used in cement, but as chemical filler.

4) Caption from Figure 1 is strange and must be modified (Figure 1. …..!). Moreover, there is not any vertical axes with the unit (it must be intensity). 

Response:Figure 1 has been modified.

5)It must be 11 A tobermorite not tobermorite 11 A.

Response:It has been modified in full text.

6) How the peaks have been distinguished for Figure 1 (is it coming from the literatures or have been analysed by you(any software) )? In case it is coming from the literatures, they must be cited.

Response:The peaks for Figure 1 were analyzed by X`pert Highscore software.

7) XRD measurement technique and its characterizations must be added to the methodology. 

Response:XRD results were analyzed by X`pert Highscore software.

8) Please make more comprehensive conclusion as in the revised version the following points must be included; materials and methods, the significant of this study, the scope of the effort, the procedures used to execute the work, and the major findings.

Response: These have been expressed in conclusion.

Round 2

Reviewer 1 Report

The authors responded well to the suggested comments; please accept the manuscript.

Author Response

Thank you very much for your valuable comments to this manuscript. 

Reviewer 2 Report

The authors focused only on linguistic suggestions, completely neglecting my major concern about the novelty of the study. I believe it is worthy of at least a response of the authors, if they disagree with my point of view (repeated below) they should argue against it.

Concerning the “Materials and Methods” and “Results” section of the manuscript. She entire approach and setup of this study is the same as several previous ones of the author group e.g. Materials 15(4),1620, Journal of Materials Research and Technology 15, pp. 4185-4192. Some parts are actually identical e.g. Table 1 of this study is transferred from a previously published work by the authors (i.e. Journal of Materials Research and Technology 15, pp. 4185-4192). In such cases authors should simply refer to the use of materials presented in the former study. The only difference of this study to the latter ones, is the evaluation of a new parameter (i.e. time) compared to temperature or C/S molar ratio, considered in the previous studies. Since no new methods or concepts approaches are introduced, I would consider this a minor contribution towards the current state of the art and label this a “case study” rather than a research article.

Author Response

Author`s reply to the review report 2 (Reviewer 2) Round 2

Concerning the “Materials and Methods” and “Results” section of the manuscript. She entire approach and setup of this study is the same as several previous ones of the author group e.g. Materials 15(4),1620, Journal of Materials Research and Technology 15, pp. 4185-4192. Some parts are actually identical e.g. Table 1 of this study is transferred from a previously published work by the authors (i.e. Journal of Materials Research and Technology 15, pp. 4185-4192). In such cases authors should simply refer to the use of materials presented in the former study. The only difference of this study to the latter ones, is the evaluation of a new parameter (i.e. time) compared to temperature or C/S molar ratio, considered in the previous studies. Since no new methods or concepts approaches are introduced, I would consider this a minor contribution towards the current state of the art and label this a “case study” rather than a research article.

Response:Thank you very much for your valuable comments to this manuscript. Although this paper and the author's previous articles published in the Journal of Materials Research and Technology and Materials both use calcium silicate slag and silica fume to hydrothermal synthesize calcium silicate materials. However, the hydrothermal synthesis in this paper is tobermorite, and more attention is paid to the growth process of tobermorite in the synthesis process. The tobermorite can be widely used in paper, plastics, rubber fillers and thermal insulation materials. The hydrothermal synthesis in other two articles are not tobermorite, so this article is essentially different from the other two articles.

Reviewer 3 Report

Dear Authors, 

- Since the introduction has lack of information about the works have been carried out earlier, please quote comments 1, 2, and 3 in a good manner as i explained earlier. It must be clear for readers to see in the manuscript what I have explained in comments 1 , 2 , and 3. 

- Caption related to Figure 1 is still strange! (The caption must be started with "Figure 1. XRD patterns of hydrothermal synthesis at different hydrothermal synthesis time ...."). I mean, why the description about A, B,C ,D,E phases place before "Figure 1. XRD patterns of hydrothermal synthesis at different hydrothermal synthesis time ...."!

- The usage of X pert Highscore software is not mentioned in the manuscript and must be cited.

Regards,

Author Response

Author`s reply to the review report (Reviewer 3) Round 2 

1.Since the introduction has lack of information about the works have been carried out earlier, please quote comments 1, 2, and 3 in a good manner as i explained earlier. It must be clear for readers to see in the manuscript what I have explained in comments 1, 2, and 3. 

Response:According to the round 1 review report, the content concerned in comments 1, 2 and 3 have been quoted in paper.

2.Caption related to Figure 1 is still strange! (The caption must be started with "Figure 1. XRD patterns of hydrothermal synthesis at different hydrothermal synthesis time ...."). I mean, why the description about A, B,C ,D,E phases place before "Figure 1. XRD patterns of hydrothermal synthesis at different hydrothermal synthesis time ...."!

Response:Figure 1 has been modified.

3.The usage of X pert Highscore software is not mentioned in the manuscript and must be cited.

Response:The X’Pert Highscore software has been cited in the paper.